# Experimental and Numerical Studies on the Behaviors of Autoclaved Aerated Concrete Panels with Insulation Boards Subjected to Wind Loading

**DOI:** 10.3390/ma14247651

**Published:** 2021-12-12

**Authors:** Junkai Lu, Jie Chen, Kun Zhu, Hang Xu, Wenjia Zhang, Qing Deng

**Affiliations:** 1School of Civil Engineering, Northeast Forestry University, Harbin 150040, China; HIT_Lujunkai@163.com; 2China Construction Science and Industry Corporation, Shenzhen 518054, China; chenjiesz@cscec.com (J.C.); xhang@cscec.com (H.X.); 3School of Civil Engineering, Changchun Institute of Technology, Changchun 130012, China; Zhangwenjia202112@163.com; 4School of Civil Engineering, Harbin Institute of Technology, Harbin 150090, China; dengqing54actor@163.com

**Keywords:** AACPs, wind suction test, numerical simulation, mechanical property, flexural behavior

## Abstract

Autoclaved aerated concrete panels (AACP) are lightweight elements in civil engineering design. In this paper, experiments and numerical analyses were conducted to study the flexural behavior of an enclosure system that consisted of AACPs and a decorative plate. A full-scale test was conducted to investigate the behavior of the enclosure system under wind suction. Load–deflection curves and load–strain relationships under different wind pressures were recorded and discussed. The effects of thickness, reinforcement ratio, and strength grade on the flexural behavior of AACPs were numerically investigated. Based on the numerical results, we found that the flexural behavior of AACPs can be improved by increasing the thickness or the reinforcement ratio. A comparison of finite element and theoretical results calculated using American and Chinese design formulae was conducted, and the results indicated the existing design formulae can conservatively estimate the major mechanical indices of AACPs.

## 1. Introduction

With the implementation of the policy for energy conservation and emission reduction, prefabricated constructions are gaining increased attention. The traditional exterior thermal insulation system usually adopts the combination of a cladding panel and a polystyrene board. This type of insulation system incurs high material costs and is prone to exterior wall detachment and burning of the construction material. Therefore, it is of great significance to develop a new assembly envelope system to solve the deficiencies of the existing system. Autoclaved aerated concrete (AAC) has the advantages of low weight, prefabrication, and good heat and sound insulation, which results in a wide prospect in engineering applications. As a lightweight element used in high-rise buildings, the wind-resistance of an enclosure system consisting of AACPs is important in engineering design, especially for buildings near coastal areas. Previous studies mainly focused on the physical properties of AAC [1,2,3]. However, there are only a handful studies focusing on the impact of wind pressure on the enclosure system.

A series of experimental tests were conducted by Nasim Uddin et al. [4] to investigate the flexural behavior of hybrid fiber-reinforced polymer–autoclaved aerated concrete panels (FRP-AAPCs). Their results showed that the flexural capacity of the composite FRP could be improved significantly. Benayoune et al. [5] tested a group of precast concrete sandwich composite panels and concluded that the flexural failure mode of the proposed composite panels was similar to that of conventional concrete panels. Carbonari et al. [6] experimentally investigated the flexural behavior of light-weight sandwich panels with perpendicular connectors under different boundary conditions and found that the shear connectors had only a small impact on the stiffness of the structure, whereas the failure mode was governed by the yield strength of rebars. Raj et al. [7] adopted basalt fiber-reinforced composite panels as the top and bottom skins of the system. The subsequent flexural tests validated the satisfactory ductility of the proposed composite panel. Amran et al. [8] conducted six full-scale tests to study the behavior of precast foamed concrete sandwich panels (PFCSPs). The results showed that the larger the length/thickness ratio, the smaller the ultimate flexural capacity of the PFCSPs. Moreover, Pan et al. [9] conducted experimental and numerical analyses of cold-formed thin-walled steel composite walls, which were infilled with light polymer material, and concluded that the steel-strength panel species had major impact on the bending capacity of the proposed composite wall. Rozylo et al. [10,11] experimentally and numerically investigated the stability of thin-walled structures made of composite materials, in which the loss of load capacity was evaluated through three independent damage models. Zang et al. [12] experimentally studied the mechanical behavior of a CFRP laminate with cutout under free vibration and numerically investigated the effects of cutout shape, size, location, and number on free vibration. It was an effective method to study the flexural performance of an AAC wall through the abovementioned static test. As AAC are one of the most commonly used enclosure systems in high-rise buildings, it is necessary to investigate the mechanical behaviors of AAC partitions and their envelope system through the wind loading test. The application of the wind suction test, which considers the dynamic response of the assembling enclosure system consisting of an AAC wall and an insulation panel, constitutes a novelty in comparison to other similar studies.

In the present study, an assembly enclosure system that consisted of an AAC wall and an insulation plate is proposed. A full-scale wind suction test was conducted to investigate the physical behavior of the envelope system, and the deflection and strain responses of the specimen were measured during the test. Finite-element analysis was carried out to study the effects of thickness, reinforcement ratio, and compressive strength on the flexural behavior of the AAC panel. Finally, a comparison of the FE and the design results, calculated with the formulae suggested by Chinese and American standards, was performed.

## 2. Materials and Methods

### 2.1. Specimens and Loading Protocols

Based on the dimensions (2890 mm × 1250 mm × 2360 mm) of the wind pressure chamber (as shown in Figure 1), a full-scale enclosure system consisting of an AAC wall and a mineral wool–aluminum plate was manufactured and tested. The AAC wall had the dimensions of 3000 mm × 150 mm × 2440 mm and consisted of four pieces of autoclaved aerated concrete panels (AACPs), whose transverse width was 600 mm. As shown in Figure 2, an HPB 300 steel rebar with a diameter of 6.5 mm was employed in the AACPs. The compressive strength of concrete was 3.23 MPa, determined using compression tests on 150 mm × 150 mm × 150 mm concrete cubes. A uniform gap of 5 mm existed between two consecutive AACPs and was filled with a masonry mortar. As shown in Figure 3a, the AAC wall was mounted in a steel frame that consisted of four rectangular steel tubes having the same dimensions through built-in anchors. The outer insulation mineral wool–aluminum plate was finally fixed on the AAC wall with the bonding mortar and anchors (Figure 3b,c).

### 2.2. Test Setup and Measurement Device

As specified in the technical guidelines for external thermal insulation engineering based on insulated decorative panels [13], the experimental procedures and requirements were set to be as follows.

1. The enclosure system specimen was closely installed on the wind pressure chamber, and a rubber insulation pad was adopted to ensure the tightness between the specimen and the chamber.

2. Based on the fundamental wind pressure of Zhanjiang city, China, three different levels of wind loads were successively applied to the full-scale specimen. A total of 4245 pulses with three stages were imposed cyclically on the specimen. A standard stage of wind load is shown in Figure 4, in which *W* denotes the design peak wind pressure. Each stage consisted of 1000, 400, 10, 4, and 1 pulses, which corresponded to 40%, 60%, 80%, 90%, and 100% of the peak pressure. Three design peak wind pressures ω of 3.50 kPa, 4.24 kPa, and 5.02 kPa, which represented the building heights of 15 m, 40 m, and 80 m (Table 1), respectively, were applied. The corresponding standard value of wind pressure can be calculated based on the load code for the design of building structures [14] as follows:(1)$$w_{k}=\beta_{\mathrm{gz}}\mu_{\mathrm{sl}}\mu_{z}w_{o}$$
in which *β*_gz_ is the gust coefficient, *μ*_sl_ denotes the local shape coefficient, *μ*_z_ means the height variation factor of wind pressure, ω_0_ is the reference wind pressure.

3. The test would be terminated if the insulated decorative panel or the anchor bolt was blown off.

Figure 5 shows different views of the test setup. A total of eight linearly variable displacement transducers (LVDTs) were settled at the back of the AAC wall to measure the lateral deformation of the panel, as shown in Figure 6. A total of 25 strain gauges were attached to the quarter heights and the panel seams and recorded the stress responses of the AAC wall and the insulation plate (Figure 7). Fourteen strain gauges were arranged at the front and back of the AAC wall, whereas the other 11 strain gauges were mounted on the outside surface of the insulation plate.

## 3. Results

As seen in Figure 8, no cracking of the decorative plate and pulling out of the connection between the AAC wall and the decorative panel were observed. Figure 9 shows the deflection curves of the W-4, W-5, and W-6 measurement points, which were located at the half-height of the AAC wall. It can be seen in Figure 9 that the deflections at W-4 were slightly greater than those at W-5 and significantly greater than those at W-6 during the test. The reason may be that the wind pressure applied on the specimen was not even, which caused the specimen to turn towards the right edge. Figure 10 presents the deflection responses of all measurement points under the highest level of wind pressure (Level 3). The maximum deflections of all measurement points were greater than 2.3 mm. However, the maximum deflection of W-6 was 1.4 mm. This indicated that a rotation around the right support was indeed caused by the wind load during the test.

Figure 11 shows the envelope strain responses at the center of the decorative panel during the last stage of the wind pressure test (Level 3). It can be drawn from Figure 11 that the strain gradually raised with the increasing of the wind suction until the peak strain was achieved. However, the strain curve in the descent stage and that in the ascent stage were not symmetric with respect to the peak point. The reason may be that plastic strain was induced with the increase of the wind load. No fracture was observed on the weatherproof sealant near the seams after the test. Moreover, the flexural strain reached the maximum value of 500 at Y-9 under the peak wind suction, which indicated that the control point was on the outer surface of the central area for the two-way panel.

## 4. Numerical Simulations

The failure mode of the assemble enclosure system was not examined due to the limited loading capacity of the wind pressure chamber, although the real responses of the enclosure system were obtained through the wind suction test. Thus, it is necessary to further investigate the mechanical behavior of the envelope system based on the finite element simulation. This section mainly focuses on the wind-resistant behavior of the AAC wall; the effect of the insulation panel was ignored due to the difficulties in accurately modeling the interaction between the built-in anchor and the panel. As the main bearing part of the enclosure system, the AAC wall can be regarded as a pure bending element under wind suction [5,7]. Therefore, static analyses in a four-point loading were adopted here.

### 4.1. Model Description

Finite element analysis was performed on the AACP using the software package ABAQUS [15]. According to the requirement of Autoclaved aerated concrete slabs [16], five different thicknesses (150 mm, 175 mm, 200 mm, 250 mm, and 300 mm), three reinforcement ratios (*ρ* = 0.2%, 0.3%, and 0.4%), and three strength grades (A3.5, A5.0, and A7.5) of concrete were adopted. Thus, a total of 18 AACP specimens were considered in the current numerical study.

All the numerical models consisted of a piece of AACP with the dimensions of 3800 mm × 600 mm × 300 mm and four rigid cushion blocks (Figure 12), and the detail of the AAC model is presented in Figure 13. The cushion blocks were connected to the AACP using the tie constraint, and the embedded region was used to simulate the interaction between the AAC and the internal rebars. Two of the cushion blocks were assembled at one quarter point of the specimen, while the others were arranged at the ends of the AACP, as shown in Figure 12. Fixed and simple constraints were adopted at the bottoms of the left and right cushion blocks, respectively, and the horizontal distance between them was 3400 mm. The lateral displacement was applied on the AAC panel through two reference points, which was coupled with the upper surface of the top cushion blocks. Both the AACP and the cushion block were modeled using eight-node brick solid elements with the reduced-integration technique (C3D8R), whereas the two-node linear truss elements (T3D2) were used for the rebars inside the concrete panel. The mesh size of 50 mm was used for the AACP and the cushion blocks, and that of 10 mm for the embedded rebars. The ‘Concrete Damaged Plasticity’ material model was adopted for the AAC. The parameters of the ‘Concrete Damaged Plasticity’ material model are listed in Table 2. The elastic perfectly plastic model with a yield stress of 350 MPa was considered for the rebar. The other material properties of AACP are presented in Table 3 [17]. Mesh analysis was conducted to consider the effect of mesh density on the simulation results. As shown in Figure 14, the flexural response of the model with a mesh density of 25 mm was almost identical to that of the model with a density of 50 mm. Thus, the mesh size of 50 mm was utilized to avoid complex calculations in the subsequent parameter analyses.

Based on the modeling strategy described above, Figure 15 and Figure 16 show the calibrated load–deflection responses and the simulated flexural behavior of model S3-A77-1 from the Chen’s experimental results [17]. The maximum error of flexural stiffness between the numerical analysis results and the test results was less than 10%, and that of ultimate flexural capacity between the two analyses was less than 8%.

### 4.2. Thickness of the AACP

In this section, the effect of the thickness of the AACP on the flexural behavior of AACPs was numerically studied. Fifteen finite element analyses were performed on the AACPs, including different thicknesses and reinforcement ratios. Three common reinforcement ratios (*ρ* = 0.2%, 0.3%, and 0.4%) were chosen for the AACP models. The moment–curvature curves of the models are shown in Figure 17. It can be seen from Figure 17 that, with the increase in thickness, the cracking moment and the ultimate flexural capacity of the AACPs increased. When the thickness exceeded the value of 200 mm, there was an apparent increase in the ultimate flexural capacity for the three groups of models. Moreover, increasing the thickness also improved the flexural stiffness of the AACPs, as seen in Figure 17.

### 4.3. Reinforcement Ratio

As mentioned in Section 4.2, three types of reinforcement ratios were considered in the current study. Figure 18 presents the moment–curvature responses of 15 numerical models. Obviously, flexural stiffness, cracking moment, and ultimate flexural capacity can be enhanced with the increase of the reinforcement ratio. Figure 19 shows the curves of cracking moment vs. thickness and ultimate flexural moment vs. thickness of finite element models. It can be seen from Figure 19 that the relationship between the cracking moment and the thickness was not linear, and neither was that between the ultimate flexural capacity and the thickness. When the thickness was increased to 200 mm, there was an apparent increase in the slope of the first curve. When the thickness reached the value of 175 mm, a slight change in the slope of the last curve was observed. Therefore, it is a cost-effective recommendation that the thickness of the AACP should be greater than 200 mm in the design to improve its flexural behavior.

### 4.4. Strength Grade of AAC

Three numerical models with different strength grades (A3.5, A5.0, and A7.5) were established to investigate the effect of the strength grade on the flexural behavior of the AACP. Figure 20 shows the moment–curvature curves of the three models. It can be seen from Figure 20 that the strength grade had little effect on the flexural behavior of the AACP. With the increase in the strength grade, the flexural stiffness of the AACP improved, whereas the cracking moment increased from 2.58 kN·m to 2.85 kN·m, indicating an enhancement of around 10.5%. The flexural capacity of AAC slab also increased from 10.16 kN·m to 10.61 kN·m, indicating an enhancement of about 4.4%. Therefore, it can be concluded that the thickness and reinforcement ratio are the main parameters affecting the flexural performance of the AACP.

### 4.5. Comparison of the Theoretical and Finite Element Results

As specified in the Chinese code for the design of concrete structures [18], the flexural capacity of concrete can be estimated using Equation (2)
(2)$$M\leq \alpha_{1}f_{c}bx\left(h_{0}-x/2\right)$$
where *b* represents the width of the AACP, *h*_0_ denotes the effective height of the cross-section of the slab, *f*_y_ is the design value of the tensile strength of the longitudinal rebar, *f*_c_ is the design value of the compressive strength of the AAC, and *A*_s_ is the cross-sectional area of the longitudinal rebar. The depth of the compression zone *x* can be obtained by solving Equation (3)
(3)$$\alpha_{1}f_{c}bx=f_{y}A_{s}$$
where α_1_ = 1.0 when the strength grade of concrete is less than C50.

According to the Chinese JGJ/T 17-2020 standard [19], the flexural capacity of the AACP can be determined using Equation (4)
(4)$$M\leq 0.75f_{c}bx(h_{o}-\frac{x}{2})$$

It is apparent that the difference between Equations (2) and (4) is the coefficient α_1_.

American RILEM [20] proposed a design correlation for calculating the flexural capacity of the AACP, as given by Equation (5)
(5)$$M=f_{c}bh^{2}\left[\alpha s\left(1-\beta s\right)+c^{,}\left(1-\frac{d_{2}}{h}\right)\right]$$
where $\alpha =1-\frac{\left(1-s\right)k}{s}\leq \alpha_{\max}=0.667$, $s=\frac{k+c-c^{\prime }}{1+k}$, $k=\frac{\varepsilon_{\mathrm{cy}}}{2\varepsilon_{\mathrm{su}}}$, $c=\frac{A_{s}f_{y}}{bhf_{c}}$, $c^{\prime }=0.75c\left(\frac{{A_{s}}^{\prime }}{A_{s}}\right)$, $\beta =\frac{2k\left(1-s\right)\left[-1+2k\left(1-s\right)/\left(3s\right)\right]+s}{2s-2k\left(1-s\right)}\leq \beta_{\max}=0.361$, *d*_2_ and *h* represent the depth of compressive and tensile rebars, respectively, *ε*_cy_ represents the maximum compressive strain of concrete, *ε*_su_ denotes the ultimate tensile strain of the rebar, and *A*_s_″ is the cross-sectional area of the longitudinal compressive rebars.

Figure 21 shows the comparison of the ultimate flexural capacities of the finite element results and the theoretical ones as calculated using the American and Chinese specifications [18,19,20]. It can be seen from Figure 21 that the three design correlations can conservatively estimate the ultimate flexural capacities of AACPs. The calculated results obtained using Equation (4) were the closest to the simulated results, with a maximum error of 27%. Moreover, Equation (4) resulted in the lowest evaluation of the ultimate flexural capacity, with a maximum error of 50%.

As specified in the Chinese code for the design of concrete structures [18], the cracking moment of concrete *M*_cr_ can be estimated using Equation (6)
(6)$$M_{\mathrm{cr}}=M_{c}+M_{s}=\frac{1}{2}f_{t}b\frac{h}{2}(h-2\times \frac{1}{3}\times \frac{h}{2})+\sigma_{s}A_{s}(h-2\alpha_{s})$$
where *E*_s_ and *E*_c_ denote the elastic moduli of the rebar and concrete, respectively, *α*_s_ represents the distance between the resultant point of tensile reinforcement and the edge, and *σ*_s_ is the stress of the rebars, which is calculated by Equation (7)
(7)$$\sigma_{s}=E_{s}\varepsilon_{s}=\frac{E_{s}}{E_{c}}\frac{h_{0}}{h}f_{t}$$

Figure 22 shows that the cracking moments of all the numerical models were greater than those obtained using Equation (6), and the maximum error between the theoretical results and the FE ones was less than 25%. Therefore, Equation (6) can be used to estimate the cracking moment of the AACP.

According to the technical specification for application of autoclaved aerated concrete [19], the flexural stiffness *B*_s_ of an AACP is expressed by Equation (8)
(8)$$B_{s}=0.85E_{c}I_{0}$$
where *I*_0_ denotes the moment of inertia of AACP. Figure 23 presents the comparison of the flexural stiffnesses calculated by the numerical modeling and the theoretical method [19]. It can be seen from Figure 23 that the maximum error between the results obtained by the two strategies was less than 30%, and the simulated results were greater than the designed ones. Therefore, Equation (8) can be used to evaluate the flexural stiffness of an AACP. This paper mainly focused on the effects of thickness, reinforcement ratio, and strength grade on the flexural behavior of AACPs. Therefore, it should be noted that the specimen parameters should be within the range selected in this paper, when referring to the results of this paper. Future concerns should be paid to the effects of concrete cover thickness and reinforcing bar strength, and the effect on the flexural behavior of AACPs should be another key issue.

## 5. Conclusions

In this study, a full-scale wind pressure test and a series of finite element analyses were conducted to investigate the flexural behavior of the assembled enclosure system. Based upon the results, following conclusions are drawn.

The assembly enclosure system consisting of the AACPs and a decorative plate exhibited good wind-resistant performance. There was no cracking of the decorative plate and no pulling out of the connection between the AAC wall and the decorative panel during the test. The control point of the decorative panel was at the center of its outside surface. The finite element results showed that the thickness of the AACP and the reinforcement ratio had major effects on the flexural behavior of the AACP, while the strength grade of AAC had little impact. The major performance indices of the AACP can be improved with the increase of thickness or of the reinforcement ratio. The current Chinese and American standards can conservatively estimate the flexural stiffness, cracking moment, and ultimate flexural capacity of an AACP. The flexural capacity of an AACP can be evaluated by the design formula proposed by American RILEM with a maximum margin of 27%. Using the equation recommended by the Chinese code for the design of concrete structures it is possible to assess the cracking moment of an AACP with a maximum error of 25%. According to the technical specification for application of autoclaved aerated concrete, the maximum error on the flexural stiffness of the AACP between the theoretical and finite element results was less than 30%.

## Figures and Tables

**Figure 1 materials-14-07651-f001:**
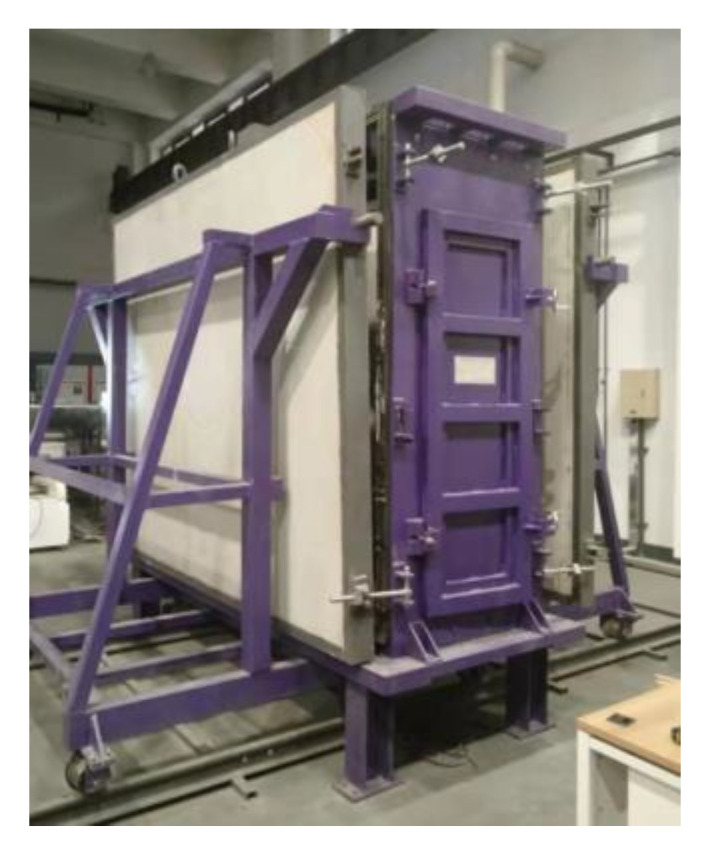
Wind pressure chamber.

**Figure 2 materials-14-07651-f002:**
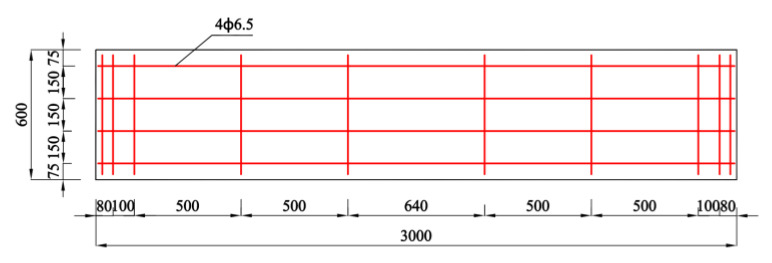
Schematic of the AACP (units: mm).

**Figure 3 materials-14-07651-f003:**
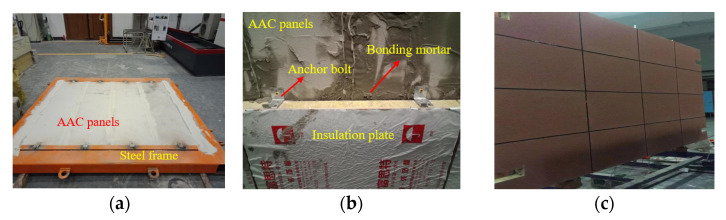
Assemblage of the enclosure system. (**a**) Assembly of AACPs and steel frame. (**b**) Assembly of AACPs and insulation plate. (**c**) Side view of the enclosure system.

**Figure 4 materials-14-07651-f004:**
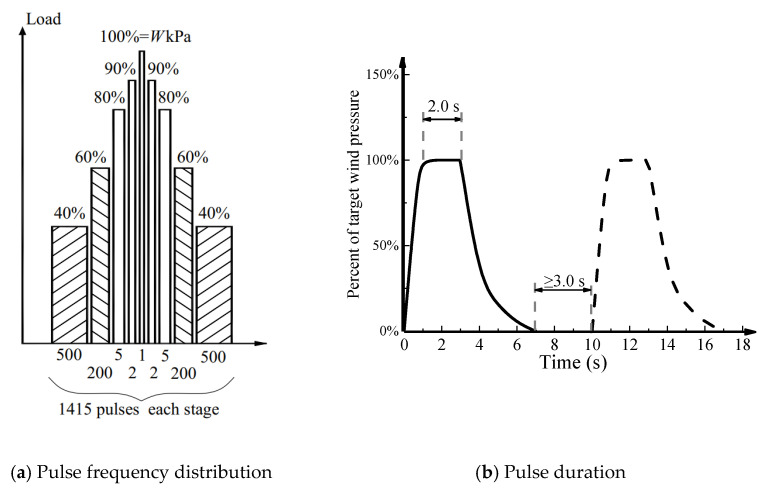
Loading protocol.

**Figure 5 materials-14-07651-f005:**
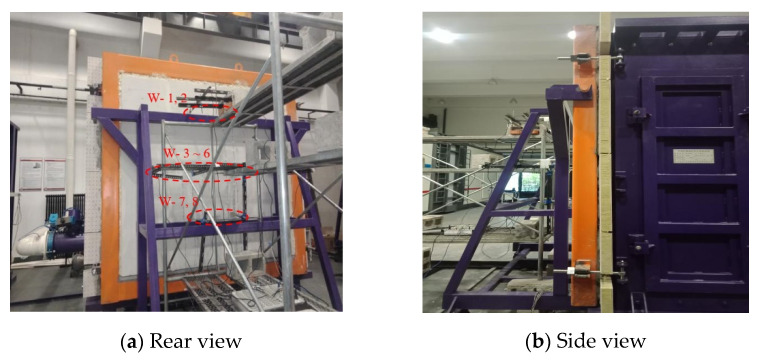
Setup for the wind suction test.

**Figure 6 materials-14-07651-f006:**
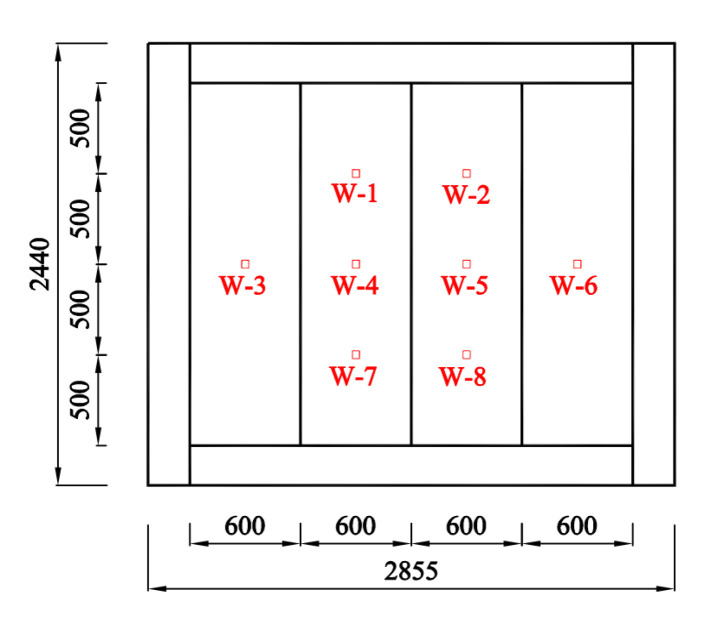
Position of the LVDT.

**Figure 7 materials-14-07651-f007:**
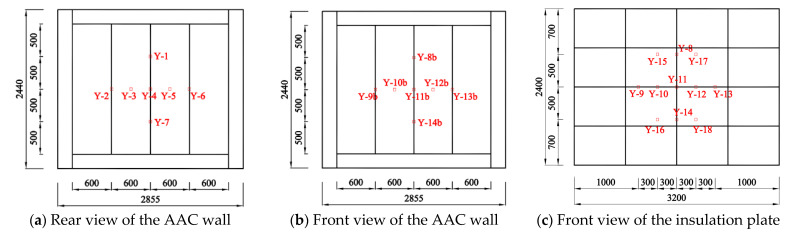
Arrangement of the gauges.

**Figure 8 materials-14-07651-f008:**
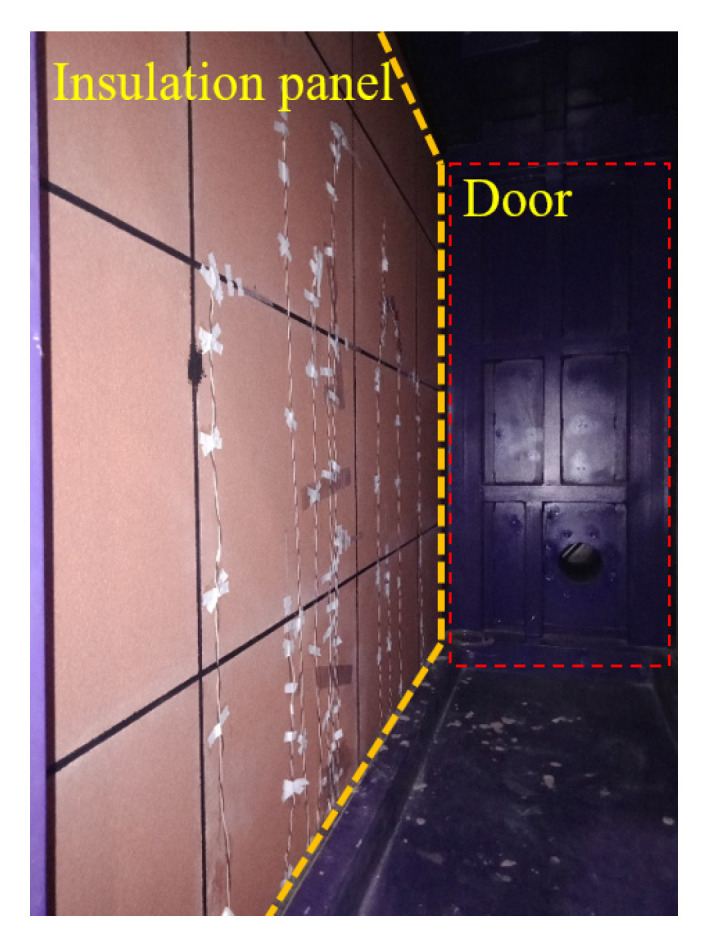
Inside view after the test.

**Figure 9 materials-14-07651-f009:**
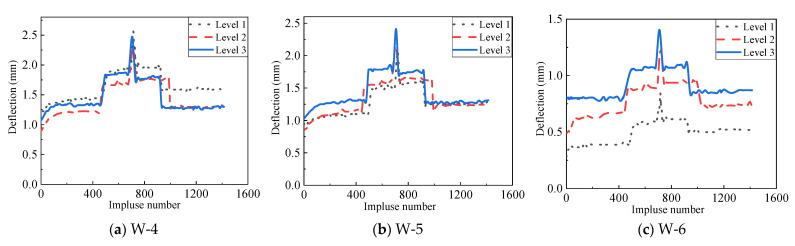
Deflection histories of the measurement points at half-height of the specimen.

**Figure 10 materials-14-07651-f010:**
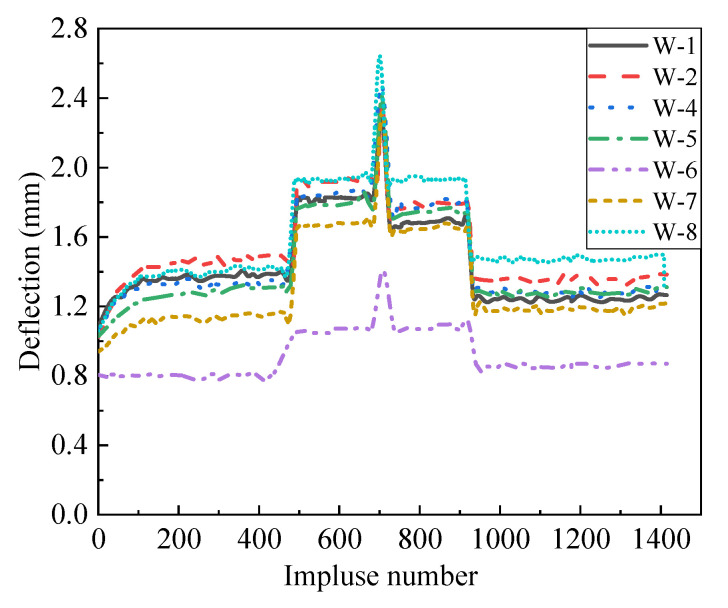
Deflection histories of all measurement points.

**Figure 11 materials-14-07651-f011:**
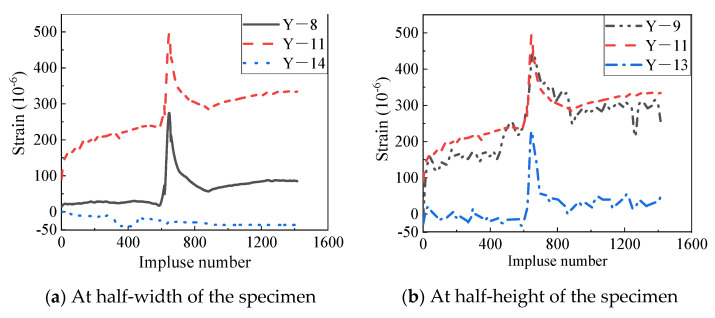
Envelope strain responses of the decorative panel.

**Figure 12 materials-14-07651-f012:**
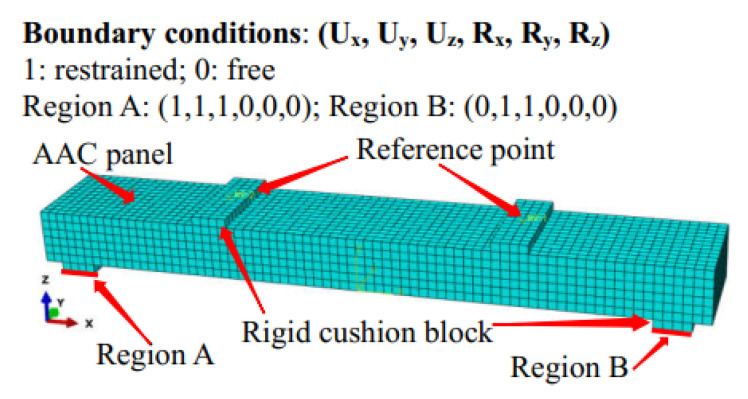
Finite element model of the specimen.

**Figure 13 materials-14-07651-f013:**
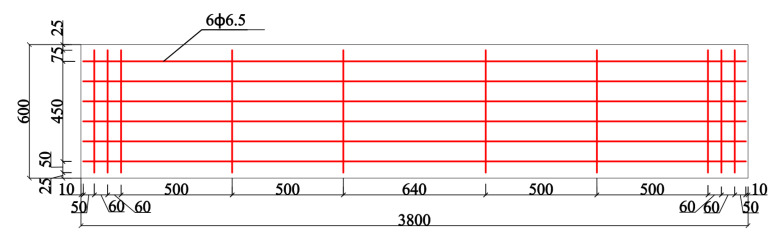
Details of the numerical model (units: mm).

**Figure 14 materials-14-07651-f014:**
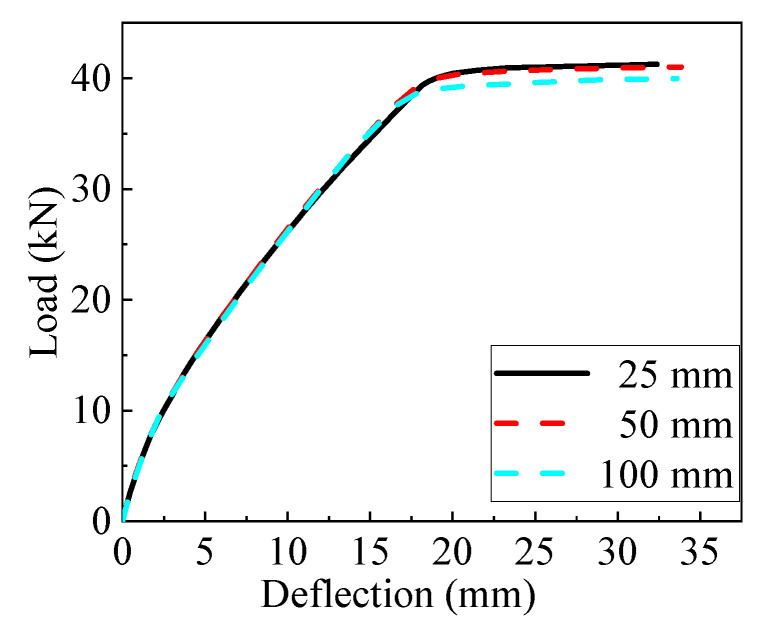
Mesh analyses results for the AACP.

**Figure 15 materials-14-07651-f015:**
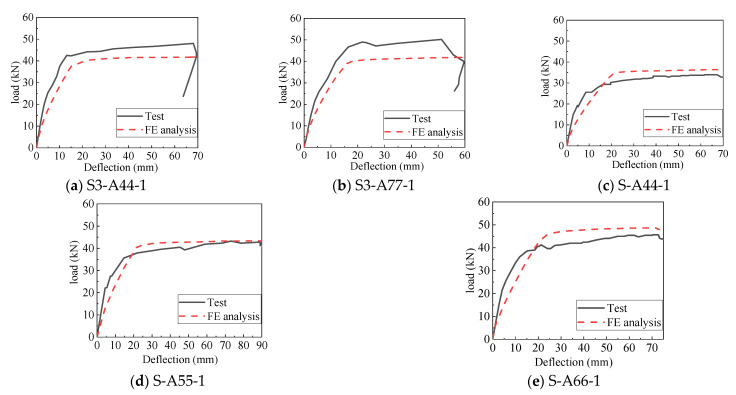
Calibrated load–deflection curves for the AACP.

**Figure 16 materials-14-07651-f016:**
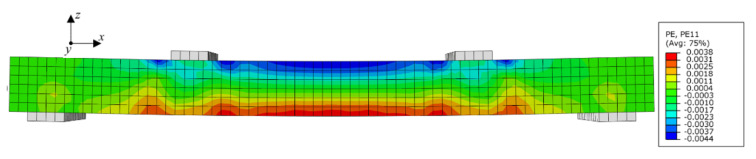
Flexural behavior of the numerical model S-A77-1 [17].

**Figure 17 materials-14-07651-f017:**
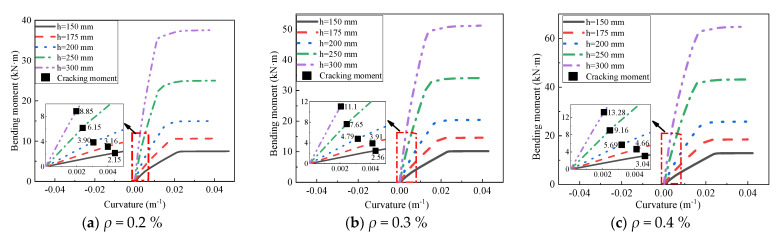
Effect of the thickness of the AAC wall.

**Figure 18 materials-14-07651-f018:**
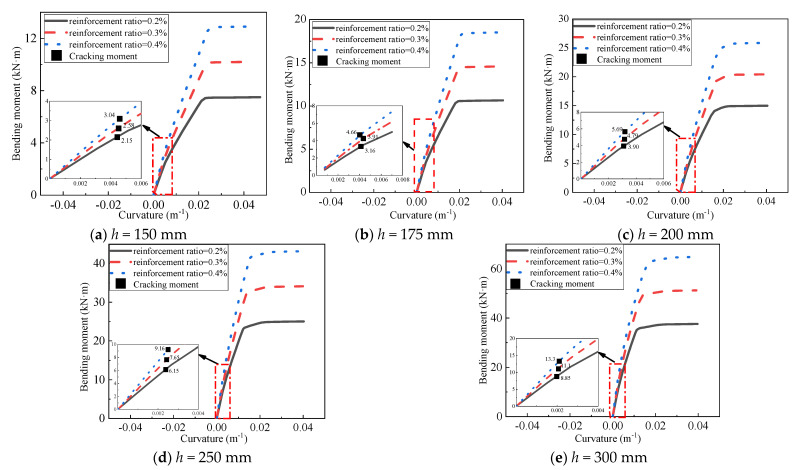
Effect of the reinforcement ratio.

**Figure 19 materials-14-07651-f019:**
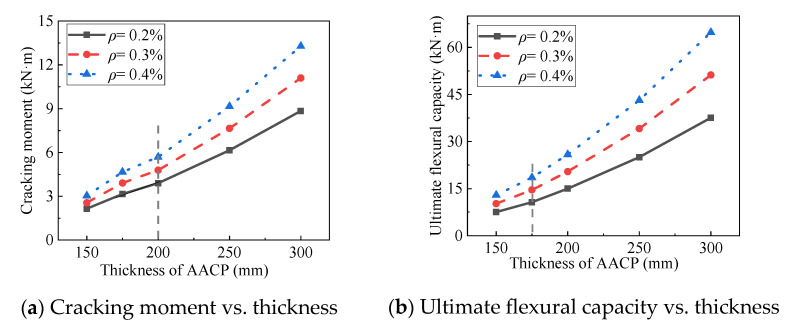
Comparison of the cracking moment (and ultimate flexural capacity) among the AACPs with different thicknesses.

**Figure 20 materials-14-07651-f020:**
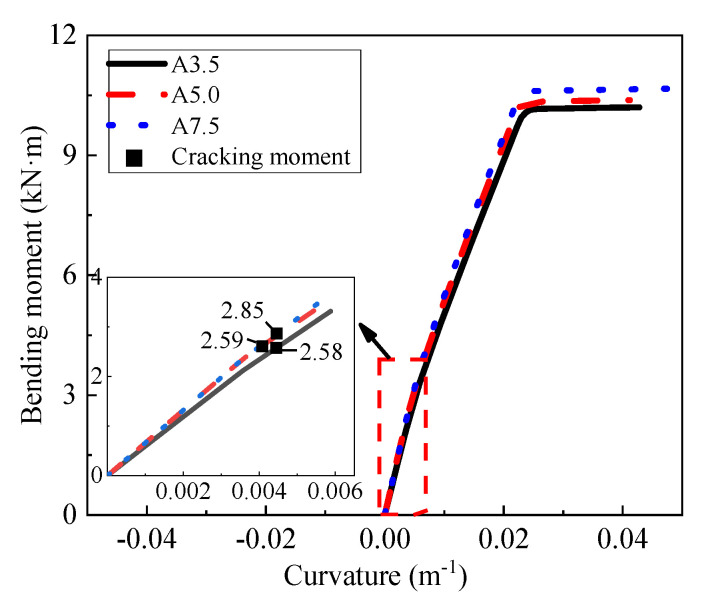
Effect of the strength grade.

**Figure 21 materials-14-07651-f021:**
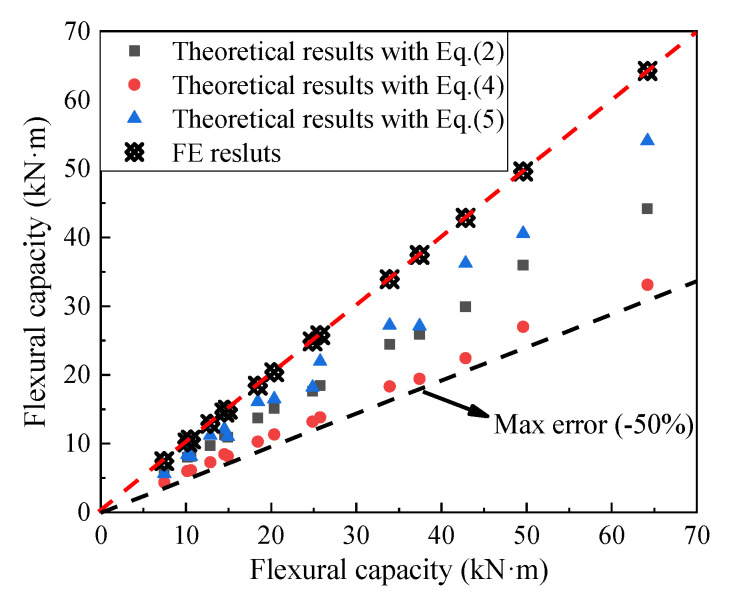
Comparison of different correlations for the flexural capacities of AACPs.

**Figure 22 materials-14-07651-f022:**
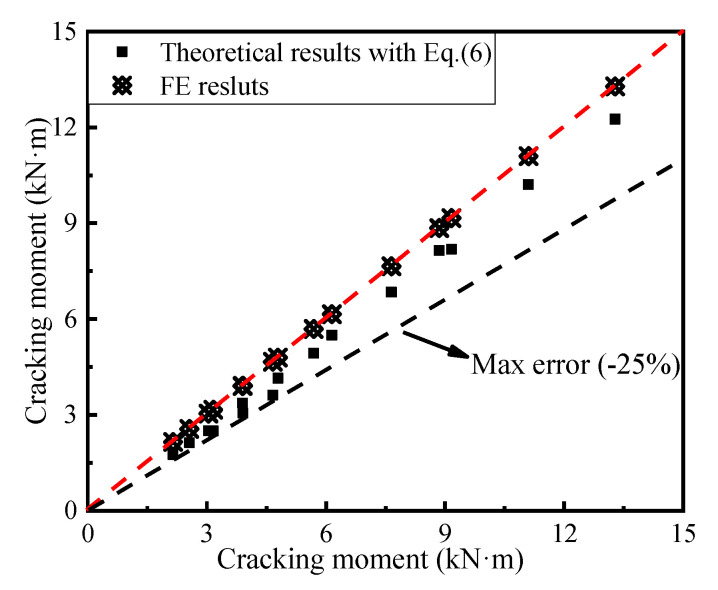
Comparison of the cracking moments calculated by the theoretical and the FE methods.

**Figure 23 materials-14-07651-f023:**
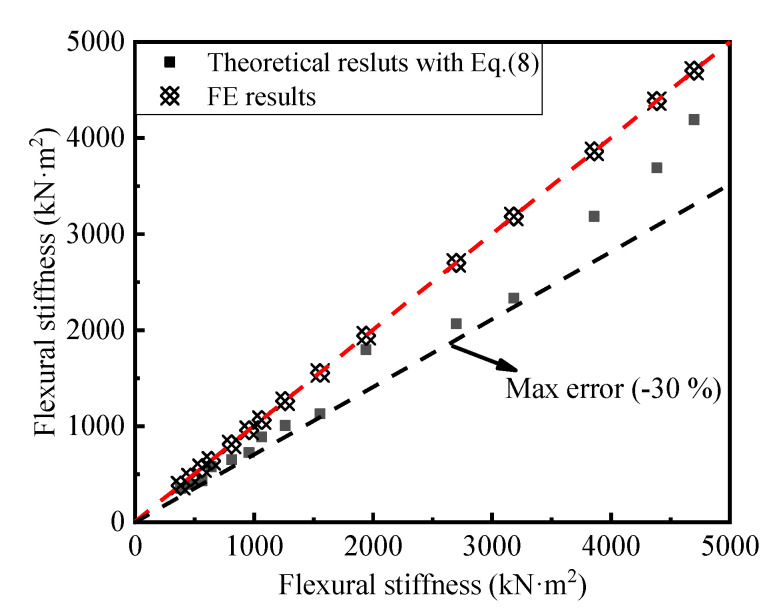
Comparison of the flexural stiffnesses.

**Table 1 materials-14-07651-t001:** Calculation coefficients corresponding to different heights.

Height (m)	*β* _gz_	*μ* _sl_	*μ* _z_	*ω* _0_	*ω*_k_ (kN/m^2^)	*ω* (kN/m^2^)
15	1.57	−1.4	1.42	0.80	−2.50	−3.50
40	1.51	−1.4	1.79	0.80	−3.03	−4.24
80	1.47	−1.4	2.18	0.80	−3.59	−5.02

**Table 2 materials-14-07651-t002:** Parameters of the concrete damaged plasticity model.

*ψ*	*ϵ*	*σ*b0/*σ*c0	*K*	*μ*
30°	0.1	1.16	0.6667	0.0005

Note: *ψ* is the dilation angle, *ϵ* is the flow potential eccentricity, *σ*_b0_/*σ*_c0_ is the ratio of the initial equibiaxial compressive yield stress to the initial uniaxial compressive yield stress, *K* is the ratio of the second stress invariant on the tensile meridian to that on the compressive meridian, and *μ* is viscosity.

**Table 3 materials-14-07651-t003:** Material properties from the coupon tests.

Category/Material Grade	Elastic Modulus (×105 MPa)	Yield Strength (MPa)	Tensile Strength (MPa)	Compressive Strength (MPa)	Poisson’s Ratio
AAC	0.02	-	0.28	2.88	0.2
Rebar	2.10	350	-	-	0.3

## Data Availability

No new data were created or analyzed in this study. Data sharing is not applicable to this article.

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
