# Peer review of "Experimental and Numerical Studies on the Behaviors of Autoclaved Aerated Concrete Panels with Insulation Boards Subjected to Wind Loading"

_materials, 2021, doi:10.3390/ma14247651_

Round 1
Reviewer 1 Report
This paper presents a very interesting and valuable issue related to the experimental and numerical study of the behavior of concrete panels subjected to wind loading. The article is presented correctly in terms of its content, but contains some shortcomings that must be corrected in order to be published:
- The literature presented in the introduction is not sufficient - the authors refer to only 9 literature references. I suggest expanding the literature a bit (10.1016/j.compstruct.2021.113598, 10.1016/j.compstruct.2021.114014)
- Please demonstrate the novelty of this paper in relation to other thematically similar scientific papers (at the introduction section).
- Please expand your conclusions to include a discussion within the quantitative evaluation of the research.
- Please be sure to correct figure 2 - it is very poor quality. The same situation applies to the figure 7.
- Has the effect of mesh density on the numerical results been studied? Please briefly describe this topic in your paper.
- It would be useful to graphically represent the boundary conditions of the numerical model.
- Why are there no results in graphical form from the numerical simulations ?
Reviewer 2 Report
Overall the manuscript was interesting. However the following can help to enhance the paper further
- The research question needs to be more clearer and explicit to the reader
- The originality and the contribution to the scientific field needs to be strengthen further.
- The limitation of the results in practical situation needs to be discussed further
- The following Figures need to be improved in terms of its presentation and quality: 2, 6, 7, 8, 15
- For the FE model, can the authors discuss more on the mesh sensitivity analysis
- Chen's experimental results were used to validate the existing model. the authors are encouraged to display through graphs the comparison between the experimental and FE model
- can the authors describe more on possible experimental errors in their study and how this was overcome.
